# IgG4 Antibodies Induced by Repeated Vaccination May Generate Immune Tolerance to the SARS-CoV-2 Spike Protein

**DOI:** 10.3390/vaccines11050991

**Published:** 2023-05-17

**Authors:** Vladimir N. Uversky, Elrashdy M. Redwan, William Makis, Alberto Rubio-Casillas

**Affiliations:** 1Department of Molecular Medicine and USF Health Byrd Alzheimer’s Research Institute, Morsani College of Medicine, University of South Florida, Tampa, FL 33612, USA; 2Biological Science Department, Faculty of Science, King Abdulaziz University, P.O. Box 80203, Jeddah 21589, Saudi Arabia; 3Therapeutic and Protective Proteins Laboratory, Protein Research Department, Genetic Engineering and Biotechnology Research Institute, City for Scientific Research and Technology Applications, New Borg EL-Arab, Alexandria 21934, Egypt; 4Cross Cancer Institute, Alberta Health Services, 11560 University Avenue, Edmonton, AB T6G 1Z2, Canada; 5Autlan Regional Hospital, Health Secretariat, Autlan 48900, Jalisco, Mexico; alberto110966@gmail.com; 6Biology Laboratory, Autlan Regional Preparatory School, University of Guadalajara, Autlan 48900, Jalisco, Mexico

**Keywords:** IgG4 antibodies, mRNA vaccines, immuno-tolerance, auto-immunity, SARS-CoV-2, COVID-19

## Abstract

Less than a year after the global emergence of the coronavirus SARS-CoV-2, a novel vaccine platform based on mRNA technology was introduced to the market. Globally, around 13.38 billion COVID-19 vaccine doses of diverse platforms have been administered. To date, 72.3% of the total population has been injected at least once with a COVID-19 vaccine. As the immunity provided by these vaccines rapidly wanes, their ability to prevent hospitalization and severe disease in individuals with comorbidities has recently been questioned, and increasing evidence has shown that, as with many other vaccines, they do not produce sterilizing immunity, allowing people to suffer frequent re-infections. Additionally, recent investigations have found abnormally high levels of IgG4 in people who were administered two or more injections of the mRNA vaccines. HIV, Malaria, and Pertussis vaccines have also been reported to induce higher-than-normal IgG4 synthesis. Overall, there are three critical factors determining the class switch to IgG4 antibodies: excessive antigen concentration, repeated vaccination, and the type of vaccine used. It has been suggested that an increase in IgG4 levels could have a protecting role by preventing immune over-activation, similar to that occurring during successful allergen-specific immunotherapy by inhibiting IgE-induced effects. However, emerging evidence suggests that the reported increase in IgG4 levels detected after repeated vaccination with the mRNA vaccines may not be a protective mechanism; rather, it constitutes an immune tolerance mechanism to the spike protein that could promote unopposed SARS-CoV2 infection and replication by suppressing natural antiviral responses. Increased IgG4 synthesis due to repeated mRNA vaccination with high antigen concentrations may also cause autoimmune diseases, and promote cancer growth and autoimmune myocarditis in susceptible individuals.

## 1. Introduction

In a relatively short period after the beginning of the COVID-19 pandemic, two mRNA vaccines, BNT162b2 (Pfizer-BioNTech, New York, NY, USA) and mRNA-1273 (Moderna, Cambridge, MA, USA), were granted the first-ever emergency use authorization. These mRNA vaccines represented a new type of vaccine that comprises synthetic mRNA molecules that contain the coding sequence necessary to build the SARS-CoV-2 Spike protein, which is encased in the lipid nanoparticles (LNPs) to allow for the delivery of mRNA to cells. The main characteristic of the mRNA vaccine platform is that the proteins are synthesized within the host cells, mimicking a natural infection with SARS-CoV-2 [1].

Contemporary investigations have contrasted the seriousness of symptoms in COVID-19 individuals infected with the SARS-CoV-2 Alpha, Delta, and Omicron variants, as well as the effectiveness of mRNA immunizations versus each variant among individuals admitted to hospitals in the United States from March 2021 to January 2022. COVID-19 vaccines were discovered to be quite efficient (90%) in avoiding intensive care unit (ICU) admissions caused by Alpha, Delta, and Omicron variants. However, three vaccine injections were needed to give protection against the Omicron variant, whereas two injections sufficiently safeguarded against the Alpha and Delta variants [2]. When people were admitted to hospitals, the Omicron variant was linked to fewer clinical adverse outcomes than the Delta variant. Despite that, the Omicron variant still produced considerable clinical symptoms and mortality [2,3,4,5,6].

It is worth noting that there are conflicting pieces of information about the level of protection offered by these vaccines. Although the Center for Disease Control (CDC) in the United States has stated that throughout the pandemic, mortality rates have been higher in the unvaccinated than in the vaccinated [7], the data in the United Kingdom contradict the CDC’s findings. Specifically, the Office for National Statistics (ONS) in the United Kingdom has reported that from April to mid-November 2021, deaths in unvaccinated people were higher in comparison with vaccinated people who had received a second vaccine dose. However, from the end of November 2021 to December 2022, this situation reverted: deaths were higher in vaccinated people who received a third vaccine dose compared with the unvaccinated [8]. Moreover, a recent work investigated a probable relationship between COVID-19 vaccination uptake in Europe in 2021 and monthly excess all-cause mortality in 2022; that is, mortality was higher than before the pandemic. All-cause mortality during the first 9 months of 2022 increased more in countries with higher 2021 vaccination uptake, according to analyses of 31 countries estimated by population size; a one percentage point increase in 2021 vaccination uptake was associated with a monthly mortality increase in 2022 of 0.105% (95% CI, 0.075–0.134). The relationship remained strong after adjusting for alternative factors [9].

Although they can induce significant neutralizing anti-spike IgG and IgA responses, all three anti-COVID-19 vaccines: Pfizer, Moderna, and Astra Zeneca ChAdOx1, (Cambridge, UK) appeared to be only transiently protective against SARS-CoV-2 infection and transmission [10,11,12,13]. The high rate of breakthrough infections brought on by the Omicron variant suggests that the sterilizing protection offered by the existing immunization schedules is minimal [14]. There are several evasion strategies that SARS-CoV-2 uses to elude immunological monitoring and attack, including the impairment of interferon synthesis [15,16,17,18,19,20], disruption in antigen presentation [21,22], evasion of humoral attack by constructing nanotubes [23,24], and induced lymphopenia through syncytia formation [25,26,27].

Lethal COVID-19 cases have been linked to higher levels of IgG4 antibodies [28,29], and it has also been documented that mRNA vaccines trigger their synthesis [30,31]. It is, therefore, important to analyze this issue in depth. In this paper, we provide the scientific rationale suggesting that repeated vaccination with mRNA vaccines could generate an immune tolerance mechanism, thereby favoring unopposed SARS-CoV-2 replication. The long-term consequence of this tolerance could be the establishment of a permissive state of the host leading to chronic infection and other unintended consequences induced by mRNA vaccination in susceptible individuals.

## 2. Characteristics of the Unusual IgG4 Antibody

Several immunoglobulin classes and subclasses that constitute the antibody immune arsenal, including IgA, IgE, IgM, and IgG, are essentially identified by the structure of their heavy chain constant region. Human immunoglobulins G (IgG) are divided into four subcategories based on the immunogenicity of their heavy chains (IgGl, IgG2, IgG3, and IgG4) [32,33,34]. Immunoglobulin subclasses differ in their basic physiologic regulation, localization throughout the organism, and engagement with receptors on immune system effector cells [35]. IgG4, the less prevalent subclass, is found in serum at mean values of 0.35–0.51 mg/mL [36], while the levels of IgG1, the most prevalent subclass, fluctuate between 5 and 12 mg/mL [37]. Due to its unusual biological characteristics and deficiency of effector functions, such as the ability to destroy infected cells through the activation of the complement system or using antibodies, IgG4 has been referred to as an unusual antibody by not adhering to the accepted theory of antibody structure and function [38,39].

The mechanism behind the reaction involving the replacement of one half of an antibody with another, also known as Fab arm exchange and specific to IgG4 antibodies, has been elucidated over the past twenty years [40]. The heavy chains can dissociate and then recombine arbitrarily due to the enhanced propensity of the natural IgG4 joint disulfide bonds to reduction, resulting in a heterogeneous group of IgG4 molecules with random heavy-chain and light-chain couples (Figure 1) [40].

The majority of IgG4 molecules will have two distinct Fab arms because of the half-antibody exchange, making them “bi-specific” and operationally univalent for a particular antigen. As a result, far from the other IgG subclasses, IgG4 antibodies in circulation are unable to form immunological complexes with antigens. IgG4 antibodies have a limited theoretical potential for immunological activation due to their weak affinity for C1q and Fc receptors. The production of immune complexes stimulates the complement system and the action of immune effector cells. Furthermore, IgG4 antibodies may be able to block the inflammatory effects of IgG1 or IgE antibodies by dislodging the binding of those with comparable specificities. The anti-inflammatory characteristic may offer insight into another important fact that IgG4 antibodies are typically formed after prolonged contact with an allergen, hence reducing the level of chronic inflammation [28].

The designation “IgG4-related systemic disease” refers to several clinical manifestations that were formerly thought to be completely distinct diseases. The list of organs linked to this illness is continuously expanding. Regardless of the organ involved, tissue biopsies show significant histological similarities. However, there are slight variations between organs as well. The hallmark pathology findings include widespread fibrosis, numerous IgG4-positive plasma cells, and disperse lympho-plasmacytoid infiltrates [42].

### 2.1. IgG4: A Protective or Pathogenic Antibody?

IgG4′s reputation as a “blocking antibody” stems from its diminished capacity to elicit immune system effector reactions [43,44]. This implies that there will only be a minimal immune response when IgG4 interacts with molecules [45]. An IgG4 response can be either pathogenic or protective, depending on the situation. For instance, IgG4 is frequently referred to as a safeguarding blocking antibody because it can suppress or halt inflammation by competing with inflammatory IgE for antigen binding in the case of allergies and infections with helminth and filarial parasites. In contrast, IgG4 can lead to serious illness in several autoimmune disorders [46] as well as cancer [47,48]. Its bi-functionality will be thoroughly examined in the next subsections.

#### 2.1.1. Protective Role of IgG4 in Allergy Immunotherapy

IgG4′s lack of effector action and the phenomena of half-antibody interchange create complicated considerations about whether these antibodies are harmful or whether they act as a counter-regulatory reaction to an enduring immunologic illness [40]. High concentrations of antigen-specific IgG4 are linked to satisfactory results in allergen-specific immunotherapy by inhibiting immunoglobulin E (IgE)-mediated effects (Figure 2), according to published studies [49,50]. In various aspects, developing a tolerance to allergens is an essential step in the development of a strong immune system. Hence, to develop prolonged desensitization against allergens, pathways involving modified allergen-specific memory T- and B-cell responses that lead to immunological tolerance are utilized [50,51,52].

#### 2.1.2. IgG4-Related Disease and Its Pathogenesis

IgG4-related disease (IgG4-RD) is a fibro-inflammatory disorder named after the presence of numerous IgG4+ plasma cells in damaged tissues and of high serum IgG4 concentrations in most, but not all, cases [53]. Several autoantibodies have been found in the serum of individuals with IgG4-RD, according to earlier reports [54,55,56,57,58,59]. Furthermore, it is well-known that steroid therapy is typically quite successful in treating IgG4-RD patients. These characteristics suggest that the illness is autoimmune in origin. Rituximab, an anti-CD20 antibody, produced remarkable clinical responses in IgG4-RD patients in recent investigations, accompanied by a sizable B cell and plasmablast decrease [60].

These results imply that increased IgG and/or IgG4 concentrations in IgG4-RD individuals may play harmful roles [61]. Because of its particular biological traits, such as the capacity to interchange Fab arms [45], the incapacity to bind complement, and the weak affinity for Fc receptors [62], IgG4 is regarded as an anti-inflammatory immunoglobulin. IgG4 antibodies do, however, function as tissue-damaging autoantibodies in some disorders, as seen in myasthenia gravis [63], idiopathic membranous glomerulonephritis [64], and pemphigus vulgaris (PV) [65].

IgG4-RD includes a “wide variety of diseases, formerly diagnosed as Mikulicz’s disease (MD) [66], autoimmune pancreatitis (AIP) [67], Riedel thyroiditis [68], interstitial pneumonitis [69,70], interstitial nephritis [71,72], prostatitis, lymphadenopathy [73,74], retroperitoneal fibrosis (RPF) [75,76], and inflammatory aortic aneurysm [77]”. It also plays a significant role in the pathogenesis of at least 13 autoimmune disorders. It has been shown that laboratory animals passively infused with human total IgG or IgG4 develop signs in 5 of these 13 disorders, proving the pathogenicity of this antibody. IgG4-induced autoimmunity is suggested by the finding that the majority of antigen-specific autoantibodies are of the IgG4 class and that their concentrations correlate with the seriousness of the sickness for the eight remaining disorders [46]. For example, Myasthenia gravis (MG), which is characterized by the production of antibodies that attach to muscle-specific kinase (MuSK), is distinguished by sporadic muscular stiffness with significant involvement of the axial and bulbar muscles. At a certain stage during the illness, a significant portion of patients requires breathing support [78,79].

After the identification of MuSK antibodies in 2001, it quickly became evident that their IgG4 subclass predominance and correlation between titers and illness severity were key findings [80,81,82]. High-purity IgG4 from MuSK MG patients was able to attach to neuromuscular connections in mouse muscle, but not IgG1–3 from the same patients or control IgG4. Injection with this antibody caused a myasthenic phenotype in immune-compromised animals [83,84,85]. These tests conclusively demonstrated IgG4 pathogenicity [86].

##### IgG4 Role in Cancer

Immune checkpoint inhibitors, often known as cancer immunotherapy agents, prevent checkpoint proteins from attaching with their associated polypeptides, allowing cytotoxic CD8+ T lymphocytes (CTLs) to attack cancer cells. Immune checkpoint-blocking (ICB) agents include anti-CTLA-4 (cytotoxic T-lymphocyte antigen 4) and anti-PD-1 (programmed cell death protein 1) monoclonal antibodies [87,88]. ICB has demonstrated therapeutic effectiveness in a wide range of cancer types, including advanced-stage cancer patients [89,90,91]. Regrettably, only 15–30% of cancer patients who have received treatment benefit from ICB’s therapeutic efficacy [92]. Most crucially, new reports show that certain cancer patients receiving anti-PD-1 monoclonal antibody treatment have rapid disease progression (also known as hyper progressive disease (HPD) instead of cancer remission [93,94,95]. Notably, the PD-1 antibody belongs to the IgG4 family. Furthermore, cancers, such as malignant melanoma [48], extrahepatic cholangiocarcinoma [96], and pancreatic cancer [97], have been linked to plasma B-cell infiltrates that are IgG4-positive. IgG4′s contribution to cancer is poorly understood, but a groundbreaking study has added important new knowledge. Karagiannis et al. [48] studied malignant melanoma and found that IL-4 and IL-10 expression was elevated and that tumor-specific IgG4 was generated locally in the tumor tissues. It is common to think of IL-10 as an anti-inflammatory cytokine; however, this is only true in low quantities, as at larger concentrations, it shows pro-inflammatory effects [98,99,100].

Karagiannis et al. [48] also found that, in contrast to cancer-specific IgG1, cancer-specific IgG4 failed to activate two immunological processes that employ antibodies to identify and destroy cancer cells. Moreover, the IgG1 antibody was able to suppress cancer progression in an in vivo model, while IgG4 failed to do so. IgG4 antibodies cannot directly attack tumor cells and can interfere with the process of tumor cell death mediated by IgG1 antibodies. The inhibition of IgG1 binding and activation by Fc RI is the mechanism behind this obstructing activity. Such findings point to a previously un-researched feature of tumor-induced immune escape: IgG4 synthesis induced by tumors limits effector immune cell activities against tumors [48]. Another work [101] came to the same conclusion; that is, the IgG4 antibody is important and necessary for cancer immune evasion. In a cohort of individuals with esophageal cancer, B cells producing high IgG4 concentrations were markedly raised in malignant cells and also high in serum samples from patients. More IgG4 seems to be linked to more aggressive cancer growth, and both were strongly associated with higher cancer malignancy and poor prognosis. It was discovered that IgG4 can contend with IgG1 (as shown in Figure 3) in binding to Fc receptors present in some immune cells in vitro. This competition results in the inhibition of typical immune responses against cancer cells, such as cell and complement cytotoxicity and cell phagocytosis, which are mediated by IgG1 antibodies.

Locally elevated levels of IgG4 in cancer tissue hindered antibody-mediated anticancer responses, assisted cancer in blocking the local immune response and indirectly aided in cancer progression. Three separate immune-potent mice models supported this theory. It was discovered that local administration of IgG4 dramatically sped up the growth of implanted colorectal and breast tumors as well as skin papillomas caused by carcinogens. Researchers also examined the IgG4 antibody Nivolumab, which is used in cancer immunotherapy, and discovered that it dramatically accelerated the development of cancer in mice when compared to phosphate buffer saline (PBS) and IgG1-treated groups [101].

Researchers used models of immunologically competent mice to evaluate their hypothesis and further explore the mechanism mediated by such antibodies. One model involved injecting non-cancer-specific IgG4 into the subcutaneous inoculation site for breast cancer cells. In comparison to other groups of mice (injected with PBS or IgG1 without IgG4), this group’s cancer cell proliferation was dramatically accelerated, generating a significantly larger cancer mass by 21 days. Because IgG4 has no direct influence on cancer cell proliferation, these findings unambiguously indicate that cancer cells utilize the IgG4 antibody to block local immunological responses and thus allow cancer growth in vivo via immune escape. This could explain the recently discovered hyper-progressive syndrome that is occasionally linked to cancer treatment with PD-1 inhibitors [101].

The immune system can detect cancers that might otherwise escape immune surveillance thanks to immune checkpoint inhibitory therapeutic antibodies that attach to the programmed cell death protein 1 (PD-1) receptor. Yet, IgG4 antibodies can also cause an autoimmune reaction by impeding the immune system’s ability to be suppressed by regulatory T cells [102]. Intriguingly, the anti-PD-1 antibodies are class IgG4, raising the concern that this therapy is a double-edged sword. For instance, patients using immune checkpoint inhibitors alone or in combination have been linked to occurrences of acute myocarditis [103,104,105,106], sometimes with lethal consequences [107].

## 3. The Role of IgG4 Antibodies Induced by Different Vaccines

An extensive review of the literature showed that mRNA vaccines are not the only ones that induce IgG4 antibody production. The HIV, Malaria, and Pertussis vaccines also elicited such a response. Overall, there are three critical factors determining the class switch to IgG4 antibodies: excessive antigen concentration, repeated vaccination, and the type of vaccine used.

### 3.1. Excessive Antigen Concentration in Vaccines

Compared to BNT162b2, the mRNA-1273 vaccine had a greater capability for inducing a prolonged IgG4 response. The amount and duration of the spike protein produced are presumably affected by the higher mRNA concentrations in the mRNA-1273 vaccine (100 µg) compared to the BNT162b2 vaccine (30 µg). Intriguingly, among the mRNA vaccines, the mRNA-1273 vaccine generated increased anti-S1 serum IgG4 concentrations in COVID-19-uninfected individuals with previously unknown repercussions on pathogen defense. Until day 270, uninfected people who received the adenovirus-based vaccine did not exhibit this long-lasting IgG4 response [31].

The problem associated with vaccines designed to be injected with a low antigen concentration is a possible absence of immunological response, and traditionally there has been a strong connection to the “more is better” school of thought that persists, especially for the wide range of infectious diseases for which there are no trustable immune predictors of vaccine-induced protection (human immunodeficiency virus (HIV), tuberculosis (TB), hepatitis C virus (HCV), etc.) [108]. A large amount (dose concentration) or repeated immunization with the same antigen (vaccine) tends to induce specific T cell tolerance (peripheral CD4) and subsequently inhibit immune responses [108,109]. However, a high antigen dose in primary immunization has been recommended for lytic infections, which is required for both humoral and cellular immunity cooperation, while a low antigen dose is recommended for boosting [110,111]. A dose escalation technique is typically employed in clinical phase I vaccine investigations to find the dose that produces the best response. While this makes sense for diseases where there is no known immunological indicator of protection (thus, a robust response is probably superior to no response), the maximum dose that was tolerated and resulted in a positive response has often been adopted for following phase II/III investigations. Yet, significant arguments against this approach are supported by several major findings [108]:

(1) When excessive quantities of antigen are injected, it can cause cell death, resulting in the loss of a specific group of T cells; this phenomenon is known as clonal deletion.

(2) Immune tolerance may develop as a result of prolonged antigen exposure. T cells are an essential part of the immune system that detects and removes infections and other foreign objects. Yet, these T cells may become desensitized and lose their capacity to react to repeated exposures when they are exposed to large concentrations of antigens, such as during repeated vaccination. Immune tolerance is a condition that can also result in the persistence of infections or the emergence of autoimmune diseases.

(3) T cells can undergo a process known as “terminal differentiation” when vaccines are given in high concentrations, at which point they become highly specialized, losing the capacity to divide and proliferate. The immune system becomes exhausted as a result and is unable to mount a successful defense against subsequent illnesses. This is a problem since it might undermine the protective advantages of vaccinations. To balance the advantages of immunological protection and the potential disadvantages of immune exhaustion, it is crucial to carefully determine the ideal dose of vaccines.

(4) Adverse outcomes are more likely to occur in groups receiving greater doses.

(5) The intensity of the reaction between an antigen and a T cell receptor or an antibody is referred to as avidity. The immune response is more effective in identifying and removing the target antigen when avidity is high. High antigen dosages, however, can result in “immune exhaustion,” a condition where the immune system’s cells become desensitized and unable to mount a successful defense. Helper T cell and antibody avidity may decline as a consequence, impairing the immunological response to the target antigen. To establish a strong and effective immune response, it is crucial to thoroughly assess the ideal antigen dosages utilized in immunotherapy [108].

Billeskov et al. [108] provided proof of cases where lower vaccine antigen doses resulted in more positive responses from T cells, both for quality as judged by several effector capabilities and preventive efficiency in both animal and human experiments, and they presented arguments for the significance of reducing antigen dose for optimum protection in some models. They also encouraged experts in T-cell vaccination, in particular, to remember that sometimes, less certainly is more. In conclusion, is there a link between antigen dose concentration, repeated exposure, and the induction of IgG4 production? Or is the elevated IgG4 concentration associated with COVID-19 vaccination due to genetic predisposition? Because approximately half of the vaccinees showed a substantial increase in IgG4 concentration after the second mRNA inoculation [30], it is evident that such an increase is not caused by a genetic predisposition. Moreover, Moderna and Pfizer used the same antigen dose for their primary and booster vaccinations, which contradicts the vaccinology paradigm showing that a low antigen dose is recommended for boosting [110,111].

### 3.2. Repeated Vaccination

#### 3.2.1. Repeated Inoculation with COVID-19 Vaccines

Researchers have reported that quickly upon the administration of the first two mRNA vaccine doses, the pro-inflammatory subclasses IgG1 and IgG3 dominated the IgG response. Nevertheless, a few months following the second Pfizer vaccine shot, spike-specific antibodies were further enhanced by a third mRNA injection and/or new infections caused by the SARS-CoV-2 variant [30]. Of all IgG antibodies generated against the spike protein, the IgG4 increased the most, steadily from 0.04% immediately after the second vaccination to 19.27% late after the third one.

Such an increase in IgG4 levels was not observed in individuals who received either the same type or a different type of SARS-CoV-2 vaccine based on adenoviral vectors, proving that, in this study, the mRNA Pfizer vaccine was the only one to cause this response. Surprisingly, 7 months after the second inoculation, the IgG4 levels in the serum of approximately half of the vaccinees surpassed the lower limit of detection [30]. To determine if the increase in IgG4 antibody concentration was exclusive to the homologous mRNA vaccination schedule utilized, researchers studied sera from an independent group that evaluated the immune system’s capacity to react to immunization schedules that are similar and different, with the Pfizer and the adenoviral vector-based vaccine from AstraZeneca. Anti-spike IgG4 antibodies were again detected in 50% of the sera from the BNT-BNT group five to six months after the second vaccination but in only one of the 51 serum samples from the other two vaccine groups. Significantly, following the third booster immunization, a significant rise in IgG4 antibody levels was detected in virtually all vaccine recipients [30].

In this regard, it was recently demonstrated that following the traditional vaccination scheme, the serum-neutralizing effectiveness in mice against the Delta and Omicron variants of the COVID-19 Pfizer vaccine was dramatically diminished after numerous booster doses [112]. Repeated antigen stimulation reportedly caused CD8+ T cells to become exhausted. These boosters also significantly diminished CD4+ and CD8+ T cell responses and enhanced programmed cell death protein 1 (PD-1) and lymphocyte activation gene-3 (LAG-3) production in these T cells [112]. Prolonged vaccination decreased the normal development of the germinal center and hindered the generation of memory B cells specific for RBD. This research additionally revealed that prolonged RBD vaccine booster immunization increased the concentration of the immunosuppressive cytokine IL-10 as well as the proportion of CD25+Foxp3+CD4+ Treg cells. The conventional SARS-CoV-2 vaccine’s ability to provide immunological protection may be significantly impacted by over-vaccination. If this happens, either newly diagnosed COVID-19 cases or people who have already contracted the virus again may have a more severe case of the illness. This concept was proposed after seeing tolerance of both the humoral and cellular immune responses to prolonged booster immunization doses [112].

#### 3.2.2. Repeated Inoculation with HIV Vaccines

A study by Chung et al. contrasted repeated immunization with similar HIV vaccines in a scenario of an HIV vaccination trial. The protection (31.2%) afforded by one vaccine (RV144) was described by the authors as being linked with the production of IgG1 and IgG3 antibodies, whereas the protection of the other vaccine (VAX003) was negligible, and was associated with the production of IgG4 antibodies after multiple rounds of vaccinations [113]. Since the VAX003 vaccine increased levels of IgG4, which have historically been linked to reduced immunological efficiency, researchers wanted to know if the IgG4 production was merely triggered in connection with a disordered functional response or if it made a significant contribution to the improperly organized response. When IgG4 antibodies were eliminated from 16 similar samples from both trials, a significant increase in ADCP activity and a tendency toward greater ADCC for the VAX003 samples in comparison to bulk IgG was observed. These findings show that IgG4 antibodies may directly decrease antibody Fc-effector function rather than only being linked to the generation of an ad hoc reaction. Compared to VAX003, which produced mono-functional antibodies with significant amounts of IgG4 following seven protein vaccinations, RV144 produced highly functional IgG3 antibodies [113]. Therefore, several vaccinations and vaccine protocols may produce persistent antibody responses, but these IgG4 antibodies may not be as effective as the IgG1 and IgG3 subclasses. As a result, the IgG subclass change from fully efficient antibodies (IgG3) to IgG4 may constitute an important obstacle to the HIV vaccine’s success [114].

Such findings are similar to those recently reported after repeated mRNA vaccination; this IgG4 class shift was linked to a decreased ability of the spike-specific antibodies to promote complement deposition and antibody-dependent cellular phagocytosis [30]. Additionally, vaccine-induced IgG3 antibodies improved immune functions such as antibody-dependent cell-mediated cytotoxicity (ADCC) and antibody-dependent cell phagocytosis (ADCP), whereas vaccine-induced IgG4 antibodies blocked these processes [113]. Similarly, in the HIV study, the removal of IgG4 antibodies from serum led to significant elevations in Fc-mediated effector activities, confirming a non-protective role for IgG4 antibodies. The unusually high production of IgG4 in the VAX003 group could be due to the repeated injection of seven vaccine doses containing a high antigen concentration in the lack of appropriate adjuvant stimulation, which may have culminated in disproportionate B cell receptor activation [113].

From these data, it is clear that IgG4 production in the VAX003 group was associated with repeated boosting (seven rounds of immunization vs. four rounds in the RV144 group), leading to reduced protection from HIV infection; moreover, this class switch to IgG4 may promote breakthrough infections due to the impairment in Fc-mediated antiviral responses [113]. This supports the notion that an increase in IgG4 subclasses could lead to extended viral persistence in case of infection, considering that Fc-mediated effector action is essential for viral elimination [30].

#### 3.2.3. Repeated Inoculation with the MALARIA Vaccine

The merozoite surface protein 1 (MSP-1), the 175-kDa erythrocyte-binding antigen (EBA-175), and the apical membrane antigen 1 (AMA-1) are the three major objectives of the natural immune response to the *Plasmodium falciparum* parasite, which causes Malaria. It was unclear, therefore, if antibodies to these antigens act as protective agents against clinical illness or only serve as exposure markers. In a group of 302 Mozambican children ages 5, 9, 12, and 24 months, highly specific tests were used to determine antibody responses to *Plasmodium falciparum* blood-stage antigens as part of a randomized, placebo-controlled trial between 2002 and 2004. The incidence of malaria throughout the follow-up period was found to be differently correlated with IgG subtype reactions to the EBA-175 antigen [115]. Since it is believed that the antibody isotype evoked by *P. falciparum* antigens is essential, the prophylactic effect of IgG has been attributed to the neutralizing (IgG1 and IgG3) rather than the non-neutralizing subtypes (IgG2) (IgG2 and IgG4) [116,117,118,119,120]. IgG1 reactivity to EBA-175 was consistent over the first year of life before rising in the following year.

While IgG4 reactivity was minimal in the first year but significantly increased by the age of 2 years, IgG3 reactivity remained moderate throughout the study period. IgG3 reactivity was stable throughout all time, while IgG4 was low during the first year but significantly increased at the age of 2 years. The study focused on the antibody responses of individuals at 5 and 12 months and investigated the incidence of malaria during two different periods of risk, from 5 to 12 months and from 12 to 24 months. In their analysis, they noticed a distinct pattern for IgG subclasses to the EBA-175 antigen: higher concentrations of particular antibodies known as neutralizing IgG1 and IgG3 were linked to a reduced likelihood of contracting malaria in the second year. As the levels of IgG1 doubled, the risk of malaria was reduced by about 50%, and when the levels of IgG3 doubled, the risk of malaria decreased by about 60% [115].

It is important to note that the probability of contracting malaria increased by around three times when non-neutralizing IgG4 levels doubled. Up to the age of 24 months, IgG1 and IgG3 demonstrated 51% and 56% protective effects, respectively; however, IgG4 was linked to a higher risk of malaria infection throughout this age range [115]. It is interesting to note that a separate study also found a link between high IgG4 levels and a higher risk of infection and malaria exacerbations [121]. This implies that IgG4 blocks the cytotoxicity of IgG2-dependent cells caused by monocytes or other effector cells. IgG4 levels and the likelihood of malaria infection were both associated with the season of malaria transmission. The fact that IgG4 concentrations significantly increased throughout the transmission season and that the rise was greater in younger individuals than in older individuals also supports an IgG4 blocking function [121]. Moreover, IgG4 has been demonstrated to prevent the opsonization of infected erythrocytes by IgG1 and IgG3 in vitro [122].

### 3.3. The Type of Vaccine Used

IgG4 responses have been infrequently reported with other vaccines, even after numerous inoculations, including that of the tetanus toxoid (TT) vaccine and the respiratory syncytial virus (RSV) [30]. These results provide support to the proposal that IgG4 class switching is not a common result of repeated antigen exposure from immunizations against other viruses or illnesses [30]. Even though natural infection with the measles virus can generate specific IgG4 antibodies [123], even persistent viral infections such as the human cytomegalovirus (HCMV) do not produce a high amount of IgG4 antibodies [124].

A recently published study found that long-term IgG4 responses were produced by the mRNA vaccines but not by the vaccines using adenoviruses [31]. It is interesting to note that two mRNA vaccines, together with one AZD1222 (AstraZeneca) inoculation with an mRNA booster, and especially the mRNA-1273 vaccine, caused prolonged anti-S1 IgG4 responses in uninfected subjects. However, researchers were unable to detect this rise after two doses of the AZD1222 vaccine in uninfected individuals up to day 270, showing that only mRNA vaccines induced detectable and prolonged IgG4 responses until day 270. Importantly, in patients who had a previous COVID-19 infection (before vaccination), IgG4 did not rise, even after mRNA injections, implying that those with higher IgG4 levels are uninfected people who were immunized with mRNA vaccines before having their COVID-19 infection [31].

Further analysis of the literature shows that only vaccines using a part of the virus produced an increase in IgG4 levels (the spike protein for the mRNA vaccines, the gp120 protein for the HIV, and the EBA-175 antigen for the malaria vaccine, respectively). Interestingly, Buhre et al. [31] found that the adenoviral vector-based vaccine from AstraZeneca did not elicit such an increase in IgG4 levels. Moreover, other studies have shown that acellular (aP) but not whole Pertussis (wP) vaccines induced IgG4 antibody production, which was also related to impaired immunity. It was demonstrated that children injected with wP vaccines had greater total and IgG1+ plasma cell responses than those injected with an aP vaccine [125]. According to results presented at the World Association for Infectious Diseases and Immune Disorders (WAidid) Congress [126], children who had received an aP vaccine at their primary immunization had significantly higher IgG4 levels than children who had received a wP vaccine. Because IgG4 antibodies are incapable of activating the complement system and, as a result, triggering antibody-dependent phagocytosis [39], it is critical for the effectiveness of a pertussis vaccine to generate a large antibodies arsenal, with IgG1 antibodies being more effective than IgG4 antibodies [126].

## 4. Discussion

Recent studies have raised concerns that inoculation with mRNA-based COVID-19 vaccines might result in the establishment of tolerance against the spike protein generated by host cells in response to vaccination. For example, a recent work by Irrgang et al. discovered that several months after the second immunization with the Pfizer vaccine, SARS-CoV-2-specific antibodies were mainly composed of non-neutralizing IgG4 antibodies, which were enhanced even more by a third mRNA vaccination and/or SARS-CoV-2 variant breakthrough infections [30]. The authors commented that “independent of the underlying mechanism, the induction of antiviral IgG4 antibodies is a phenomenon infrequently described and raises important questions about its functional consequences” [30]. IgG4 antibodies are bi-functional: they can be protective but can also be directly pathogenic [127]. There has been a lot of research on IgG4 in chronic allergen exposure models, where natural immunological tolerance is induced by giving an allergen in increasing doses [128]. The increase in IgG4 levels after the third immunization with the Pfizer vaccine could reflect a tolerance mechanism that could prevent immune over-reactivity (cytokine storm) and progression to a critical stage [30]. However, this exacerbated immune reaction does not occur in young and healthy people, and it has been documented only in older patients with genetic susceptibility and those with comorbidities [129].

It has been suggested that an increase in IgG4 levels could have a protective role similar to that occurring during successful allergen-specific immunotherapy by inhibiting IgE-induced effects [30]. Allergen tolerance is an immune system adaptation characterized by a particular non-inflammatory response to an allergen that, under other conditions, would probably result in cell-mediated or humoral immunity, which would cause tissue inflammation and/or IgE synthesis [128]. In other words, the immune system “learns” to tolerate a foreign, although innocuous, antigen. However, a very different situation occurs when a virus invades our body. In this scenario, vaccine-induced tolerance can potentially have several negative, unintended consequences because tolerance to the spike protein could inhibit the immune system from detecting and attacking the pathogen (Figure 4); thus, potentially exacerbating SARS-CoV2 pathology in susceptible individuals who suffer re-infection of COVID-19 in the setting of vaccine-induced immune suppression. For example, it was demonstrated that patients with severe COVID-19 who passed away had higher IgG4 levels than those who recovered [28]. More precisely, the death rate increased noticeably at 30 days when serum IgG4 concentrations were above 700 mg/dL, and the ratio of IgG4 to IgG1 was above 0.05 [29]. Moreover, IgG4 levels were correlated with IL-6 levels [130], a known determinant of COVID-19-related mortality [130,131,132].

This leads us to conclude that it is incorrect to compare the increase in IgG4 levels between allergy treatments and the reported increase in IgG4 antibodies after repeated vaccination or infection with SARS-CoV-2. The induced tolerance against the spike protein could produce an impaired immune response against the virus when these patients suffer a re-infection. Although the new Omicron subvariants have a high rate of transmissibility, the severity of infections has fortunately been reduced as a result of a change in affinity towards the upper respiratory tract [27,133,134,135]. These findings may explain why Omicron infections caused fewer severe effects [136,137]. However, without an adequate protection level, even the new Omicron sub-variants (considered as mild) could cause severe multi-organ damage and death in immuno-compromised individuals and those with comorbidities.

A study by Gazit et al. found that when the initial event (infection or vaccination) happened during January and February of 2021, SARS-CoV-2-naive vaccinees exhibited a 13.06-fold (95% confidence interval (CI), 8.08–21.11) greater risk for breakthrough infection with the Delta variant compared to the unvaccinated-previously-infected persons. The increased risk for symptomatic illness was also substantial. Evidence of waning naturally generated immunity was shown when the infection happened at any point between March 2020 and February 2021, albeit SARS-CoV-2 naive vaccinees still had a 5.96-fold (95% CI: 4.85–7.33) higher risk of breakthrough infection and a 7.13-fold (95% CI: 5.51–9.21) higher risk of symptomatic disease. This research also showed that immunity acquired through natural disease provides better protection against infection and disease symptoms caused by the Delta variant of SARS-CoV-2 than the immunity provided by two injections with the BNT162b2 vaccine [138].

Even the protection that COVID-19 vaccines provide against severe symptoms and hospitalization is now being questioned following an outbreak in an Israeli hospital that resulted in the deaths of five individuals (all with comorbidities) who were fully immunized [138]. This study casts some doubt on the notion that widespread immunization will produce herd immunity and stop COVID-19 outbreaks. This may have been true for the SARS-CoV-2 wild-type virus, but in the outbreak that is the subject of the cited study, 96.2% of those who were exposed received full vaccinations [139]. Similarly, Brosh-Nissimov et al. reported that among 17 Israeli hospitals, 34/152 (22%) fully immunized patients passed away from COVID-19. Noticeably, these individuals had a high prevalence of co-morbid disorders, such as congestive heart failure, chronic renal insufficiency, high blood pressure, diabetes, and lung disorders, that made them more vulnerable to developing severe COVID-19 [140].

Irrgang et al. [30] reported that it takes months for the IgG4 class switch to develop. Could this increase in IgG4 levels explain the reduced efficacy of mRNA vaccines detected after 6 months [141]? Based on findings from the HIV trial [113], where decreased vaccine efficacy was linked to IgG4 production, we conclude that repeated mRNA vaccination is also correlated with reduced efficacy in protecting people from re-infection due to an increase in IgG4 levels.

There is now compelling evidence that, among COVID-19 vaccines, only the mRNA vaccines (but not the adenoviral vector-based vaccine from AstraZeneca) induced a remarkable increase in IgG4 levels, and such an increase was detected in SARS-CoV-2 uninfected individuals who received mRNA vaccinations before becoming infected with the virus, whereas for patients who had a previous infection before vaccination, IgG4 levels did not rise [31]. This is in contrast with findings from another study showing that the highest IgG4 levels were found in those individuals who developed a breakthrough infection after receiving three doses of mRNA vaccination, indicating that SARS-CoV-2 infections can also induce IgG4 production [30]. We suggest more research is needed for a definitive conclusion about these different results.

The HIV [113] and malaria trials [115], and studies with the Pertussis vaccine informed us that repeated vaccination was linked to reduced protection from infection, and this poor response was directly related to a higher IgG4 production. Moreover, it was suggested that this class switch might contribute to breakthrough infections due to impaired fc-mediated antiviral responses [113]. All in all, reviewed data indicate that IgG4 production induced by repeated vaccination does not in any way constitute a protective mechanism. There are also warning signs in recent literature that indicate the cellular immune response induced by the typical vaccination course may be severely compromised by repeated administration of the same booster shot or infection following vaccination, which, in combination with impaired antibody immune responses, may cause recipients’ symptoms to worsen or their disease to last longer. Excessive vaccination is likely to create an immunosuppressive microenvironment that is crucial for promoting immunological tolerance. These findings show that repeated SARS-CoV-2 booster immunization in dense populations should be approached with caution [112].

We propose a hypothetical immune tolerance mechanism induced by mRNA vaccines, which could have at least six negative unintended consequences:

(1) By ignoring the spike protein synthesized as a consequence of vaccination, the host immune system may become vulnerable to re-infection with the new Omicron subvariants, allowing for free replication of the virus once a re-infection takes place. In this situation, we suggest that even these less pathogenic Omicron subvariants could cause significant harm and even death in individuals with comorbidities and immuno-compromised conditions.

(2) mRNA and inactivated vaccines temporally impair interferon signaling [142,143], possibly causing immune suppression and leaving the individual in a vulnerable situation against any other pathogen. In addition, this immune suppression could allow the re-activation of latent viral, bacterial, or fungal infections and might also allow the uncontrolled growth of cancer cells [144].

(3) A tolerant immune system might allow SARS-CoV-2 persistence in the host and promote the establishment of a chronic infection, similar to that generated by the hepatitis B virus (HBV), the human immune deficiency virus (HIV), and the hepatitis C virus (HCV) [145].

(4) The combined immune suppression (produced by SARS-CoV-2 infection [15,16,17,18,19,20,21,22] and further enhanced by vaccination [142,143,144]) could explain a plethora of autoimmune conditions, such as cancers, re-infections, and deaths temporally associated with both. It is conceivable that the excess deaths reported in several highly COVID-19-vaccinated countries may be explained, in part, by this combined immunosuppressive effect.

(5) Repeated vaccination could also lead to auto-immunity: in 2009, the results of an important study went largely unnoticed. Researchers discovered that in mice that are otherwise not susceptible to spontaneous autoimmune disorders, repeated administration of the antigen promotes systemic autoimmunity. The development of CD4+ T cells that can induce autoantibodies (autoantibody-inducing CD4+ T cells, or aiCD4+ T cells), which had their T cell receptors (TCR) modified, was triggered by excessive stimulation of CD4+ T cells. The aiCD4+ T cell was generated by new genetic TCR modification rather than a cross-reaction. The excessively stimulated CD8+ T cells induced them to develop into cytotoxic T lymphocytes (CTL) that are specific for an antigen. These CTLs were able to mature further by antigen cross-presentation, so in that situation, they induced autoimmune tissue damage resembling systemic lupus erythematosus (SLE) [146]. According to the self-organized criticality theory, when the immune system of the host is continually overstimulated by antigen exposure at concentrations higher than the immune system’s self-organized criticality can tolerate, systemic autoimmunity inevitably occurs [147].

It has been proposed that the amount and duration of the spike protein produced are presumably affected by the higher mRNA concentrations in the mRNA-1273 vaccine (100 µg) compared to the BNT162b2 vaccine (30 µg) [31]. Thus, it is probable that the spike protein produced in response to mRNA vaccination is too high and lasts too long in the body. That could overwhelm the capacity of the immune system, leading to autoimmunity [146,147]. Indeed, several investigations have found that COVID-19 immunization is associated with the development of autoimmune responses [148,149,150,151,152,153,154,155,156,157,158,159,160,161,162,163,164,165,166].

(6) Increased IgG4 levels induced by repeated vaccination could lead to autoimmune myocarditis; it has been suggested that IgG4 antibodies can also cause an autoimmune reaction by impeding the immune system’s ability to be suppressed by regulatory T cells [102]. Patients using immune checkpoint inhibitors alone or in combination have been linked to occurrences of acute myocarditis [103,104,105,106,107], sometimes with lethal consequences [102]. As anti-PD-1 antibodies are class IgG4, and these antibodies are also induced by repeated vaccination, it is plausible to suggest that excessive vaccination could be associated with the occurrence of an increased number of myocarditis cases and sudden cardiac deaths.

Finally, these negative outcomes are not expected to affect all people who have received these mRNA vaccines. Individuals with genetic susceptibility, immune deficiencies, and comorbidities are probably the most likely to be affected. However, this gives rise to a disturbing paradox—if people who are the most affected by the COVID-19 disease (the elderly, diabetics, hypertensive, and immunocompromised people like those with HIV) are also more susceptible to suffering the negative effects of repeated mRNA vaccination, is it then justified to booster them? As Omicron subvariants have been demonstrated to be less pathogenic [133,134,135,136,137], and mRNA vaccines do not protect against re-infection [14,138], clinicians should be aware of the possible detrimental effects on the immune system by administering boosters.

## Figures and Tables

**Figure 1 vaccines-11-00991-f001:**
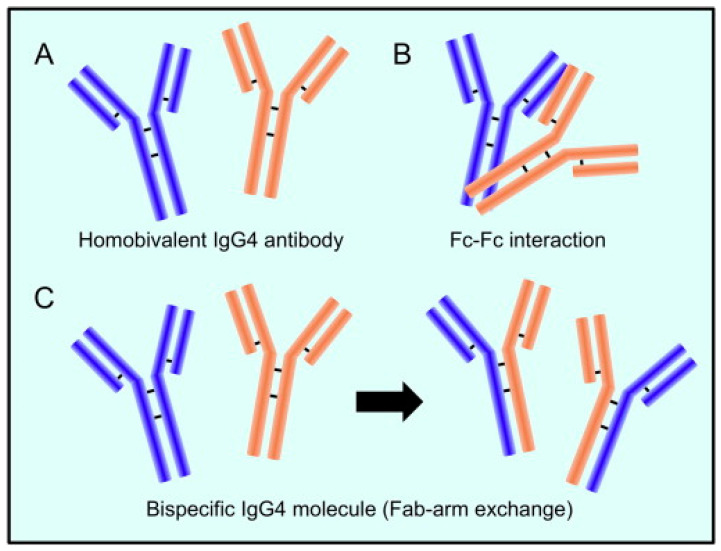
IgG4 antibody has a distinctive structure. (**A**). Two heavy chains and two light chains make up the IgG4 antibody. (**B**). The Fc fragment of one IgG4 molecule can react with the Fc fragment of another. (**C**). When half-molecules are exchanged (called a Fab-arm interchange), IgG4 combines two distinct specificities into a unique molecule (bispecific antibody). Reproduced from [41]. This is an open-access article distributed under the terms of the Creative Commons CC-BY license, which permits unrestricted use, distribution, and reproduction in any medium, provided the original work is properly cited.

**Figure 2 vaccines-11-00991-f002:**
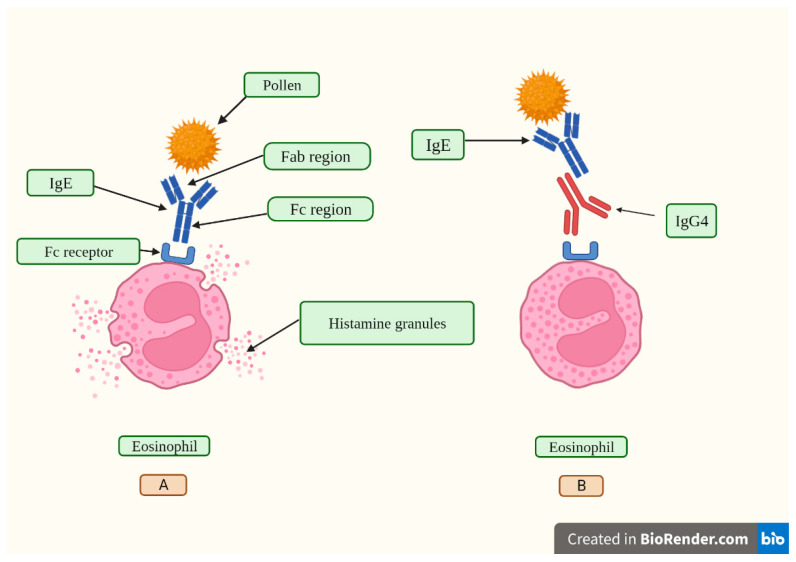
In (**A**), a pollen grain is recognized through the fragment antigen-binding region (Fab) of an IgE antibody. After that, the IgE attaches to its receptor, called Fc epsilon RI (FcεRI), located on eosinophil leukocytes, and induces histamine release from cytoplasmic granules. Histamine is a vasoactive peptide that causes symptoms such as itching, sneezing, runny nose, itchy throat, eyes, and ears, and trouble breathing during a pollen-induced allergic reaction. In (**B**), the fragment cristalizable (Fc) region of an IgG4 antibody binds to the Fc region of an IgE antibody, inhibiting its binding to the FcεRI receptor and thus blocking IgE-mediated effects. Created with Biorender.

**Figure 3 vaccines-11-00991-f003:**
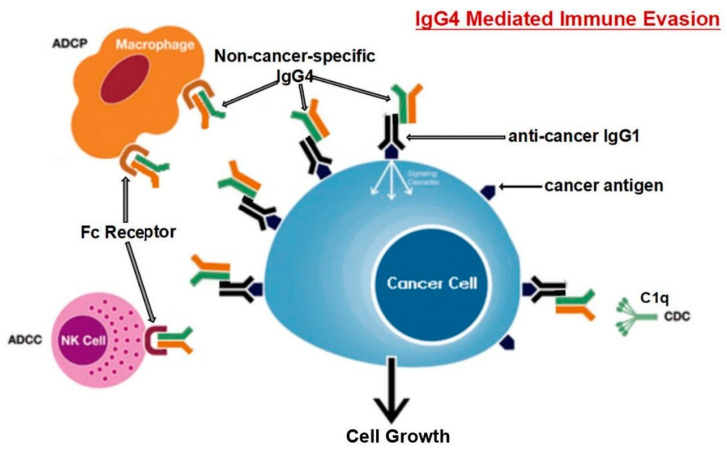
The suggested pathway for immune evasion evolved by cancer cells through IgG4 produced from B lymphocytes is depicted diagrammatically. Prolonged exposure to cancer antigens causes B cells to change their class and generate IgG4. With its Fc-Fc binding characteristic, such enhanced IgG4 can interact with cancer-bound IgG as well as Fc receptors on immune effector cells. Increased IgG4 in the cancer microenvironment promotes an efficient immune evasion mechanism for cancer due to its special structural and biological properties. The acronyms ADCC, ADCP, CDC, and NK stand for antibody-dependent cell-mediated cytotoxicity, antibody-dependent cell phagocytosis, complement-dependent cytotoxicity, and natural killer cells, respectively. Reproduced from [101]. This is an open-access article distributed under the Creative Commons Attribution Non-Commercial (CC BY-NC 4.0) license, which permits others to distribute, remix, adapt, build upon this work non-commercially, and license their derivative works on different terms, provided the original work is properly cited, appropriate credit is given, any changes made indicated, and the use is non-commercial.

**Figure 4 vaccines-11-00991-f004:**
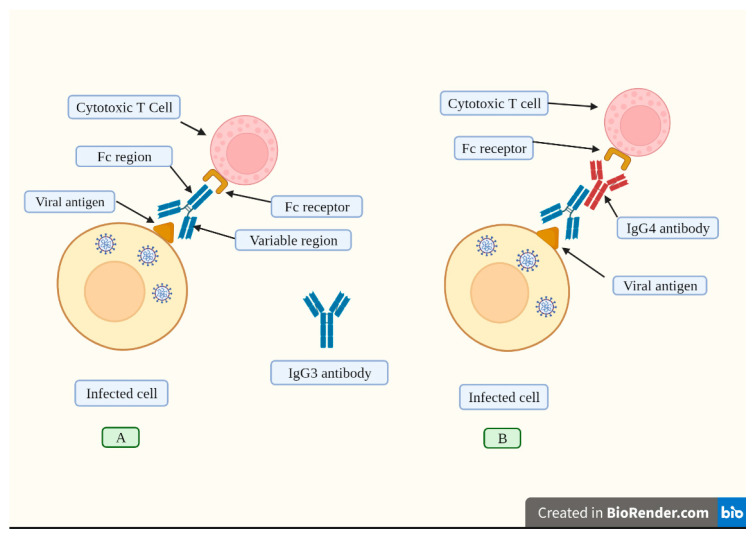
An effective humoral response induced by vaccination consists of the synthesis of high IgG3 concentrations. (**A**). IgG3 antibodies attach to viral antigens exposed on infected cells’ membranes through its variable region. This antibody has a constant region (Fc) that is recognized by the corresponding receptor found on cytotoxic T cells and other immune cells. The cytotoxic T cell becomes activated and releases chemical agents that destroy the infected cell. (**B**). Repeated vaccination induces high IgG4 levels (depicted in red). This antibody inhibits the attachment of the Fc region from the IgG3 antibody to its receptor located on cytotoxic T cells, thus blocking its activation, and consequently, the infected cell is not destroyed. In this sense, repeated boosting causes a switch to the production of high IgG4 levels, which impair immune responses. Created with Biorender.

## Data Availability

Not applicable.

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
