# Peer review of "IgG4 Antibodies Induced by Repeated Vaccination May Generate Immune Tolerance to the SARS-CoV-2 Spike Protein"

_vaccines, 2023, doi:10.3390/vaccines11050991_

Round 1

Reviewer 1 Report

The manuscript is well written and really interesting.

I suggest authors to discuss the fact that mRNA vaccines have been the choice for children, pregnant women, and immunosupressed patients in many countries based on response from clinical trials. Were IgG4 levels after mRNA investigated on these groups?

I strongly recommend discussing the possible mechanisms that trigger the IgG to IgG4 class switch by mRNA stimulation.

Author Response

Reviewer #1:

The manuscript is well-written and really interesting.

I suggest authors to discuss the fact that mRNA vaccines have been the choice for children, pregnant women, and immunosuppressed patients in many countries based on response from clinical trials. Were IgG4 levels after mRNA investigated on these groups?

Response: We thank reviewer #1 form his(her) important comments. As far as we know, the levels of IgG4 have not been investigated in these groups after mRNA vaccination.

I strongly recommend discussing the possible mechanisms that trigger the IgG to IgG4 class switch by mRNA stimulation.

Response: Although our work is a hypothesis proposal, we performed a search in the literature and found that the increased IgG4 synthesis is not exclusive from mRNA vaccines. It was also reported in the HIV and malaria clinical trials. Importantly, we found that 3 factors determine the class switch to IgG4 antibodies:

  1. Excessive antigen concentration in vaccines (page 8)
  2. Repeated vaccination with mRNA vaccines, HIV and Malaria vaccines (pages 9-11).
  3. The type of vaccine used (page 12):

Only vaccines using a part of the virus produced an increase in IgG4 levels (the spike protein for the mRNA-based vaccines, the gp120 protein for the HIV, and the erythrocyte binding antigen (EBA-175) for the malaria vaccine, respectively). Interestingly, Buhre et al (ref. 31) found that the adenoviral vector-based vaccine from AstraZeneca did not elicit such an increase in the IgG4 levels.  Other studies showed that acellular (aP) but not whole pertussis (wP) vaccines induced IgG4 antibody production, which was also related to impaired immunity.  Diks et al. showed that the total and IgG1+ plasma cell responses were stronger in subjects primed with wP vaccines than in individuals primed with aP vaccine [125]. Consistently with these data, results showing significantly higher IgG4 levels in children who had received an aP vaccine at primary immunization than those in children who had received a wP vaccine were presented at the World Association for Infectious Diseases and Immune Disorders (WAidid) Congress [126]. IgG4 antibodies are not capable of activating the complement system, and consequently initiating antibody-dependent phagocytosis [39], and because of that, it is critical for the efficacy of a pertussis vaccine to induce a broad repertoire of antibodies where induction of IgG1 antibodies being superior to IgG4 subclass [126].

Reviewer 2 Report

IgG4 antibodies induced by mRNA vaccines generate immune tolerance to SARS-CoV-2´spike protein by suppressing the immune system” (Vaccines-2334542)

The style of this manuscript fulfills more like a review article than a hypothesis. Most of all, the hypothesis that “IgG4 antibodies induced by mRNA vaccines…” was not supported by their collected references. In the introduction, the author(s) started with the invention of mRNA vaccines, which contain synthetic mRNA for SARS-CoV-2 spike protein. The artificial mRNA was wrapped into lipid nanoparticles for injection delivery and cell entry. Among all the globally available vaccines, there are only two brands of mRNA vaccines, which the authors claimed in this manuscript responsible for the low efficacy in defending against SARS-CoV-2 pandemic. They stated that the ineptness of the vaccines could be due to the repeated mRNA vaccination, which would probably induce the production of IgG4 to suppress patient’s immune system, and render patients vulnerable to SARS-CoV-2 infection, breakthrough infection, or re-infection. They then went on with the characteristics of IgG4 and IgG4-related diseases. Surprisingly, they described that the failure of vaccine development against HIV and malaria was due to the increased serum levels of IgG4, but these two examples are not mRNA vaccines. These examples cannot support their title statement that “IgG4 antibodies induced by mRNA vaccines …”. In conclusion section, the authors further itemized six possible negative consequences of mRNA vaccines, and repeated on their point of view on the disease progression of SARS-CoV-2 infection, and the lack of benefit to the COVID-19 patients.

This study would be more valuable, if more coherent description and discussion were properly organized in the manuscript. If possible, arrange a meta-analysis of some existed medical data, and get a real confirmation of the conclusion.

Major Suggestions:

1.      Language: The title needs some rearrangement or modification to show scientific merit. “IgG4 antibodies induced by mRNA vaccines“ in the title may not be appropriate. SARS-CoV-2´” (with an extra ‘) in the title should be checked. The context of the manuscript contains a number of misused words, repetitious writing as well as grammatical and logic errors. The manuscript needs English editing.

2.      Concept and data presentation:

(1)      Questionable concept: (A) Authors might need to fully understand the mechanism in class switch of IgG4 antibody production in B cells before jump to the conclusion too early. Take vaccines against HIV and those against malaria as examples, it is quite clear that neither of those two kinds of vaccine is constituted of mRNA, but of pox viral or adenoviral vector or of recombinant proteins per se, and they provoke production of IgG4 as well (page 10, lessons from the HIV vaccine trial and page 11, lessons from the Malaria vaccine trial). (B) So were hepatitis virus B and hepatitis virus C, in which the immune tolerance is not built up by high dosage of viral titer, but congenital infection through umbilical cord from the pregnant mother prior to the establishment of fetus’ own immune system. (page 14, lines 595-597). (C) Difference of IgG4 levels between patients infected by measles virus (a single-stranded, negative-sense, enveloped RNA virus) or those by human cytomegalovirus (Beta-Herpesviridae family with enveloped double-stranded DNA) has further suggested that the induction mechanism of IgG4 biosynthesis would be more sophisticated than what was proposed in this manuscript – repetitious vaccination with mRNA agents (page 8, lines 308-314).

(2)      Requires more adequate data presentation. (A) Since the relationship between increased IgG4 levels and repeated SARS-CoV-2 mRNA vaccination has been shown in several published papers (Della-Torre, 2021; Irrgang, 2022; Buhre, 2023), it would be interesting to still categorize this manuscript as hypothesis. Moreover, even the Figures are all borrowed from other sources. If possible, a meta-analysis of some existed medical data would add more confirmation of the conclusion. (B) The authors should broaden their view and knowledge of true science. There are at least five different vaccines available, and ChAdOX1, with chimp adenovirus as vector for spike gene, from Astra Zeneca was least effective. Novavax, an oligopeptide vaccine, is also from USA. The effect of Novavax vaccine should resemble that of Malaria vaccine. [The World Health Organization (WHO) recommends widespread use of the RTS,S/AS01 (RTS,S) malaria vaccine among children in sub-Saharan Africa and in other regions with moderate to high P. falciparum malaria transmission. The recommendation is based on results from an ongoing pilot program in Ghana, Kenya and Malawi that has reached more than 900 000 children since 2019. Four injection doses of vaccine are recommended.] (C) How about the role of antigen-presenting cells in those infected patients in determining the progressive stages of disease as well as their influence on survival rate? One of the antigen-presenting cells, in addition to the intermediate T cells, plays a more significant role in antibody-dependent cellular phagocytosis (Figure 3). The same cells also play an important role in HIV infection. (D) How about the pathological findings of bone marrow in these patients? Previous studies had demonstrated that SARS-CoV-2 infection did not affect bone marrow. If that is true, then there should be a reasonable explanation for the decrease of peripheral immunophenotypic cells destined for the terminal differentiation in the essential lymph nodes.     

I suggest that the manuscript should have a proof for English editing prior to submission. The authors may focus more on antigen dosage and total numbers of vaccination (injection) in COVID-19 vaccines and other vaccines and rewrite the review, instead of emphasizing “IgG4 antibodies induced by mRNA vaccines”, which has no sufficient data to prove the hypothesis.

The title needs some rearrangement or modification to show scientific merit. “IgG4 antibodies induced by mRNA vaccines“ in the title may not be appropriate. SARS-CoV-2´” (with an extra ‘) in the title should be checked. The context of the manuscript contains a number of misused words, repetitious writing as well as grammatical and logic errors. The manuscript needs English editing.

Author Response

Reviewer #2:

“IgG4 antibodies induced by mRNA vaccines generate immune tolerance to SARS-CoV-2´spike protein by suppressing the immune system” (Vaccines-2334542)

Comments and Suggestions for Authors

The style of this manuscript fulfills more like a review article than a hypothesis. Most of all, the hypothesis that “IgG4 antibodies induced by mRNA vaccines…” was not supported by their collected references.

Response: We thank reviewer # 2 for his (her) insightful comments.  However, we respectfully disagree with this statement: “the hypothesis is not supported by their collected references”. There are several experimental studies demonstrating a IgG4-induced immune supression.

As an example, in page 10, lines 420-433:

“As a result, the IgG subclass change from the fully efficient antibodies (IgG3) to IgG4 may constitute an important obstacle on the way to the HIV vaccine success [114].

Such findings are similar to those recently reported after repeated mRNA vaccination; this IgG4 class shift was linked to a decreased ability of the spike-specific antibodies to promote complement deposition and antibody-dependent cellular phagocytosis [30]. Additionally, vaccine-induced IgG3 antibodies improved immune functions such as antibody-dependent cell-mediated cytotoxicity (ADCC) and antibody-dependent cell phagocytosis (ADCP), whereas vaccine-induced IgG4 antibodies blocked these processes [113]. Similarly, in the HIV study, the removal of IgG4 antibodies from serum led to the significant elevations in the Fc-mediated effector activities, confirming a non-protective role for IgG4 antibodies. The unusually high production of IgG4 in the VAX003 group could be due to the repeated injection of seven vaccine doses containing high antigen concentration in the lack of appropriate adjuvant stimulation, which may have culminated in disproportionate B cell receptor activation [113].

From these data, it is clear that the IgG4 increase associated with repeated vaccination represents an immune suppression mechanism because IgG4 antibodies impair antiviral responses such as ADCC and ADCP.

In the case of the Malaria vaccine, we wrote:    Moreover, IgG4 has been demonstrated to prevent the opsonization of infected erythrocytes by IgG1 and IgG3 in vitro [122].

Thus, the presented evidence is sufficient to demonstrate that the vaccine-induced increase IgG4 is associated with an immune suppressive mechanism.

 In the introduction, the author(s) started with the invention of mRNA vaccines, which contain synthetic mRNA for SARS-CoV-2 spike protein. The artificial mRNA was wrapped into lipid nanoparticles for injection delivery and cell entry. Among all the globally available vaccines, there are only two brands of mRNA vaccines, which the authors claimed in this manuscript responsible for the low efficacy in defending against SARS-CoV-2 pandemic. They stated that the ineptness of the vaccines could be due to the repeated mRNA vaccination, which would probably induce the production of IgG4 to suppress patient’s immune system, and render patients vulnerable to SARS-CoV-2 infection, breakthrough infection, or re-infection. They then went on with the characteristics of IgG4 and IgG4-related diseases. Surprisingly, they described that the failure of vaccine development against HIV and malaria was due to the increased serum levels of IgG4, but these two examples are not mRNA vaccines. These examples cannot support their title statement that “IgG4 antibodies induced by mRNA vaccines …”.

Response: We sincerely thank reviewer # 2 for this correction. We have changed the title to read:

IgG4 antibodies induced by repeated vaccination may generate immune tolerance to the SARS-CoV-2spike protein.

Please note that we have used the word may, which emphasizes the hypothetical nature of our work. This will help to avoid misinterpretations.

In conclusion section, the authors further itemized six possible negative consequences of mRNA vaccines, and repeated on their point of view on the disease progression of SARS-CoV-2 infection, and the lack of benefit to the COVID-19 patients.

This study would be more valuable, if more coherent description and discussion were properly organized in the manuscript. If possible, arrange a meta-analysis of some existed medical data, and get a real confirmation of the conclusion.

Response : Thank you very much for this important suggestion. We have rewritten that section as follows. Please note we have been very careful by writing words such as might, may, and could, which denote possibility.  We also added a clarifying sentence that emphasizes our proposal is hypothetical:

Our work must be read and interpreted for what it is: a hypothesis, which must be experimentally evaluated to be confirmed or refuted. The proposed immune tolerance mechanism induced by mRNA vaccines could have at least 6 negative unintended consequences:

  • By ignoring the spike protein synthesized as a consequence of vaccination, the host immune system may become vulnerable to re-infection with the new Omicron subvariants, allowing for free replication of the virus once a re-infection takes place. In this situation, we propose that even these less pathogenic Omicron subvariants could cause significant harm and even death in immuno-compromised individuals and patients with comorbidities.
  • mRNA and inactivated vaccines temporally impair interferon signaling [143,144], possibly causing immune suppression and leaving the individual in a vulnerable state against any other pathogen. In addition, this immune suppression could allow the re-activation of latent viral, bacterial, or fungal infections, and might also allow uncontrolled growth of cancer cells [145].
  • A tolerant immune system might allow SARS-CoV-2 persistence in the host and promote the establishment of a chronic infection, similar to that generated by the hepatitis B virus (HBV), the human immune deficiency virus (HIV), and the hepatitis C virus (HCV) [146].
  • The combined immune suppression (produced by SARS-CoV-2 infection and further enhanced by vaccination) could explain a plethora of autoimmune conditions, as well as cancers, re-infections, and deaths temporally associated with both. It is conceivable that the excess deaths reported in several highly COVID-19-vaccinated countries may be explained, in part, by this combined immunosuppressive effect.
  • Repeated vaccination could also lead to auto-immunity. In 2009, the results of an important study went largely unnoticed. Researchers discovered that in mice that are otherwise not susceptible to spontaneous autoimmune disorders, repeated administration of the antigen promotes systemic autoimmunity. The development of CD4+ T cells that can induce autoantibodies (autoantibody-inducing CD4+ T cells, or aiCD4+ T cells), which had their T cell receptors (TCR) modified, was triggered by excessive stimulation of CD4+ T cells. The aiCD4+ T cell was generated by new genetic TCR modification rather than a cross-reaction. The excessively stimulated CD8+ T cells induced them to develop into cytotoxic T lymphocytes (CTL) that are specific for an antigen. These CTLs were able to mature further by antigen cross-presentation, so in that situation, they induced autoimmune tissue damage resembling systemic lupus erythematosus (SLE) [147].According to the self-organized criticality theory, when the immune system of the host is continually overstimulated by antigen exposure at concentrations that are higher (see page 11) than the immune system's self-organized criticality can tolerate, systemic autoimmunity inevitably occurs [148]. It has been proposed that the amount and duration of the spike protein produced are presumably affected by the higher mRNA concentrations in the mRNA-1273 vaccine (100 µg) compared to the BNT162b2 vaccine (30 µg) [31]. Thus, it is probable that the spike protein produced in response to mRNA vaccination is too high and last too much time in the body. That overwhelms the capacity of the immune system, thus leading to autoimmunity [147,148]. Indeed, several investigations have found that COVID-19 immunization is associated with the development of autoimmune responses [149-167].
  • Increased IgG4 levels induced by repeated vaccination could lead to autoimmune myocarditis: It has been suggested that IgG4 antibodies can also cause an autoimmune reaction by impeding the immune system`s ability to be suppressed by regulatory T cells [102]. Patients using immune checkpoint inhibitors alone or in combination have been linked to occurrences of acute myocarditis [103-107], sometimes with lethal consequences [102]. As anti-PD-1 antibodies are class IgG4, and these antibodies are also induced by repeated vaccination, it is plausible to suggest that excessive vaccination could be associated with the occurrence of an increased number of myocarditis cases and sudden cardiac deaths.

Major Suggestions:

  1. Language: The title needs some rearrangement or modification to show scientific merit. “IgG4 antibodies induced by mRNA vaccines“ in the title may not be appropriate. “SARS-CoV-2´” (with an extra ‘) in the title should be checked. The context of the manuscript contains a number of misused words, repetitious writing as well as grammatical and logic errors. The manuscript needs English editing.

Response: We thank reviewer for these important suggestions. We tried our best to corrected linguistic isues. To this end, a native English speaker was invited to edit the revised version of the manuscript.

  1. Concept and data presentation:

(1)      Questionable concept: (A) Authors might need to fully understand the mechanism in class switch of IgG4 antibody production in B cells before jump to the conclusion too early.

Response: We have cited published experimental findings from the HIV trial:

“The unusually high production of IgG4 in the VAX003 group could be due to the repeated injection of seven vaccine doses containing high antigen concentration in the lack of appropriate adjuvant stimulation, which may have culminated in disproportionate B cell receptor activation”.

Take vaccines against HIV and those against malaria as examples, it is quite clear that neither of those two kinds of vaccine is constituted of mRNA, but of pox viral or adenoviral vector or of recombinant proteins per se, and they provoke production of IgG4 as well (page 10, lessons from the HIV vaccine trial and page 11, lessons from the Malaria vaccine trial).

Response : We sincerely thank reviewer # 2 for this correction. To address this critique, we have changed the title to read:

IgG4 antibodies induced by repeated vaccination may generate immune tolerance to the SARS-CoV-2spike protein.

We also added more discussion of these important points

(B) So were hepatitis virus B and hepatitis virus C, in which the immune tolerance is not built up by high dosage of viral titer, but congenital infection through umbilical cord from the pregnant mother prior to the establishment of fetus’ own immune system. (page 14, lines 595-597).

Response: We thank for this important suggestion. As we commented before, this is a proposal and it should be experimentally tested.  Please note we now wrote:

A tolerant immune system might allow SARS-CoV-2 persistence in the host and promote the establishment of a chronic infection.

Indeed, this is according to the suggestion made by Irrgang et al:  “ This supports the notion that an increase in IgG4 subclasses could lead to extended viral persistence in case of infection, considering that Fc-mediated effector action is essential for viral elimination [30]. 

(C) Difference of IgG4 levels between patients infected by measles virus (a single-stranded, negative-sense, enveloped RNA virus) or those by human cytomegalovirus (Beta-Herpesviridae family with enveloped double-stranded DNA) has further suggested that the induction mechanism of IgG4 biosynthesis would be more sophisticated than what was proposed in this manuscript – repetitious vaccination with mRNA agents (page 8, lines 308-314).

Response: We thank reviewer for this clarification. However, we cited these studies because they were also cited by Irrgang et al.

(2)      Requires more adequate data presentation. (A) Since the relationship between increased IgG4 levels and repeated SARS-CoV-2 mRNA vaccination has been shown in several published papers (Della-Torre, 2021; Irrgang, 2022; Buhre, 2023), it would be interesting to still categorize this manuscript as hypothesis.

Response : We have clarified our work is a hypothesis. Please see page 15. If reviewer agree, we can write this clarification also in the abstract.

Moreover, even the Figures are all borrowed from other sources. If possible, a meta-analysis of some existed medical data would add more confirmation of the conclusion. 

Response: In fact, only 2 figures are borrowed from other authors, and we properly acknowledged their contribution. The other 2 figures are original.

(B) The authors should broaden their view and knowledge of true science. There are at least five different vaccines available, and ChAdOX1, with chimp adenovirus as vector for spike gene, from Astra Zeneca was least effective. Novavax, an oligopeptide vaccine, is also from USA. The effect of Novavax vaccine should resemble that of Malaria vaccine. [The World Health Organization (WHO) recommends widespread use of the RTS,S/AS01 (RTS,S) malaria vaccine among children in sub-Saharan Africa and in other regions with moderate to high P. falciparum malaria transmission. The recommendation is based on results from an ongoing pilot program in Ghana, Kenya and Malawi that has reached more than 900 000 children since 2019. Four injection doses of vaccine are recommended.]

Response: We are quoting published experimental works who have demonstrated that: “In their analysis, they noticed a distinct pattern for IgG subclasses to the EBA-175 antigen: higher concentrations of particular antibodies known as neutralizing IgG1 and IgG3 were linked to a reduced likelihood of contracting malaria in the second year. As the levels of IgG1 doubled, the risk of malaria was reduced by about 50%, and when the levels of IgG3 doubled, the risk of malaria decreased by about 60% [115].

It is important to note that the probability of contracting malaria increased by around three times when non-neutralizing IgG4 levels doubled. Up to the age of 24 months, IgG1 and IgG3 demonstrated 51% and 56% protective effects respectively, however, IgG4 was linked to a higher risk of malaria infection throughout this age range [115]. It's interesting to note that a separate study also found a link between high IgG4 levels and a higher risk of infection and malaria exacerbations. [121]. This implies that IgG4 blocks the cytotoxicity of IgG2-dependent cells caused by monocytes or other effector cells. IgG4 levels and likelihood of a malaria infection were both associated with the season of malaria transmission. The fact that IgG4 concentrations significantly increased throughout the transmission season and that the rise was greater in younger individuals than in older individuals also supports an IgG4 blocking function [121]. Moreover, IgG4 has been demonstrated to prevent the opsonization of infected erythrocytes by IgG1 and IgG3 in vitro [122].

So, we are confident that we have cited “true science”.

(C) How about the role of antigen-presenting cells in those infected patients in determining the progressive stages of disease as well as their influence on survival rate? One of the antigen-presenting cells, in addition to the intermediate T cells, plays a more significant role in antibody-dependent cellular phagocytosis (Figure 3).

Response: We do not understand what the reviewer is talking about. Figure 3 is related to the suggested pathway for immune evasion evolved by cancer cells through IgG4 produced from B lymphocytes

The same cells also play an important role in HIV infection. (D) How about the pathological findings of bone marrow in these patients? Previous studies had demonstrated that SARS-CoV-2 infection did not affect bone marrow. If that is true, then there should be a reasonable explanation for the decrease of peripheral immunophenotypic cells destined for the terminal differentiation in the essential lymph nodes.     

Response: We sincerely think that this issue, although important, goes beyond the scope of this work.

I suggest that the manuscript should have a proof for English editing prior to submission. The authors may focus more on antigen dosage and total numbers of vaccination (injection) in COVID-19 vaccines and other vaccines and rewrite the review, instead of emphasizing “IgG4 antibodies induced by mRNA vaccines”, which has no sufficient data to prove the hypothesis.

Response: We thank reviewer for this important suggestion. The manuscript was edited by native English speaker. We have also reorganized our manuscript according to the excellent comments and recommendations of this reviewer.

Round 2

Reviewer 2 Report

The major problem with this manuscript is that it is not a hypothesis. Because all data are from others' reports, and the authors did not provide their own experimental data. Therefore, it is more like a review. Previous studies about other viruses have pointed out the phenomenon. To categorize this manuscript as a hypothesis would give the reader a false perception that they are the first to propose the idea. 

Author Response

Reviewer # 2:

The major problem with this manuscript is that it is not a hypothesis. Because all data are from others' reports, and the authors did not provide their own experimental data. Therefore, it is more like a review. Previous studies about other viruses have pointed out the phenomenon. To categorize this manuscript as a hypothesis would give the reader a false perception that they are the first to propose the idea".

R: We sincerely thank reviewer 2 for his (her) pertinent comments. We have reviewed our work again and accept that our article should be classified as a review, since as you correctly mentioned that all data are from others' reports.

In page 1 (top), We have changed the word hypothesis for review. And in page 15, we have deleted this text: Our work must be read and interpreted for what it is: a hypothesis, which must be experimentally evaluated to be confirmed or refuted.

We have added this text after the deleted one: Here, we are proposing a hypothetical immune tolerance mechanism induced by mRNA vaccines, which could have at least 6 negative unintended consequences:

We hope that the changes made meet the expectations of reviewer #2 and our work is accepted for publication

Sincerely,

Alberto Rubio Casillas

Round 3

Reviewer 2 Report

In the title, “IgG4 antibodies induced by repeated vaccination may generate immune tolerance to the SARS-CoV-2spike protein”, a space should be inserted between SARS-CoV2 and spike. The correct title should be “IgG4 antibodies induced by repeated vaccination may generate immune tolerance to the SARS-CoV-2 spike protein”.

Line 284-285, “raising the concern this therapy is a double edge sword” should be “raising the concern that this therapy is a double edge sword”.

Line 291-293, “Furthermore, we found that 3 factors determine the class switch to IgG4 antibodies…..used.” should be “Overall, there are three critical factors determining the class switch to IgG4 antibodies….used [30-31, 39, 108-126].”. 

Line 627, “we are proposing a hypothetical …..” should be “we propose a hypothetical .….”.

Line 635, “…..in individuals with comorbidities and immuno-compromised.” should be “…..in individuals with comorbidities and immuno-compromised condition.”

The quality of English is acceptable. However, I found a few minor errors in the manuscript (as listed in my comments). I suggest that You have an English editing expert to check one more time.

Author Response

Dear Reviewer # 2: We sincerely thank you for the excellent corrections/suggestions you have made to our work. Your contribution will surely improve the quality of the final version. 

We have added to the text all the final suggestions you made, so, we sincerely wish to thank both reviewers for all the time and intellectual effort dedicated to the analysis of our work.

Best regards, 

Alberto Rubio Casillas